# StableMoFusion: Towards Robust and Efficient Diffusion-based Motion Generation Framework

## ABSTRACT

Thanks to the powerful generative capacity of diffusion models, recent years have witnessed rapid progress in human motion generation. Existing diffusion-based methods employ disparate network architectures and training strategies. The effect of the design of each component is still unclear. In addition, the iterative denoising process consumes considerable computational overhead, which is prohibitive for real-time scenarios such as virtual characters and humanoid robots. For this reason, we first conduct a comprehensive investigation into network architectures, training strategies, and inference processs. Based on the profound analysis, we tailor each component for efficient high-quality human motion generation. Despite the promising performance, the tailored model still suffers from foot skating which is an ubiquitous issue in diffusion-based solutions. To eliminate footskate, we identify foot-ground contact and correct foot motions along the denoising process. By organically combining these well-designed components together, we present StableMoFusion, a robust and efficient framework for human motion generation. Extensive experimental results show that our StableMoFusion performs favorably against current state-of-the-art methods.

## CCS CONCEPTS

• **Computing methodologies** → *Artificial intelligence*; **Animation**.

## KEYWORDS

Human Motion Generation, Diffusion Model, Efficient Inference, Footskate Cleanup

## 1 INTRODUCTION

Human motion generation aims to generate natural, realistic, and diverse human motions, which could be used for animating virtual characters or manipulating humanoid robots to imitate vivid and rich human movements without long-time manual motion modeling and professional skills[1, 4, 36]. It shows great potential in the fields of animation, video games, film production, human-robot interaction and *etc*. Recently, the application of diffusion models to human motion generation has led to significant improvements in the quality of generated motions [3, 27, 36].

**Unpublished working draft. Not for distribution.**

Permission to make digital or hard copies of all or part of this work for personal or classroom use is granted without fee provided that copies are not made or distributed for profit or commercial advantage and that copies bear this notice and the full citation on the first page. Copyrights for components of this work owned by others than the author(s) must be honored. Abstracting with credit is permitted. To copy otherwise, or republish, to post on servers or to redistribute to lists, requires prior specific permission and/or a fee. Request permissions from permissions@acm.org.

*ACM MM, 2024, Melbourne, Australia*

© 2024 Copyright held by the owner/author(s). Publication rights licensed to ACM.

ACM ISBN 978-x-xxxx-xxxx-x/YY/MM

https://doi.org/10.1145/nnnnnnn.nnnnnnn

Table 1: StableMoFusion achieves superior performance on motion generation compared to other state-of-the-art methods. Lower FID and higher R Precision mean, the better.

| Method | FID↓ | R Precision (top3)↑ |
|---|---|---|
| MDM [27] | 0.544 | 0.611 |
| MLD [3] | 0.473 | 0.772 |
| MotionDiffuse [36] | 0.630 | 0.782 |
| ReMoDiffuse [37] | 0.103 | 0.795 |
| StableMoFusion (Ours) | 0.098 | 0.841 |

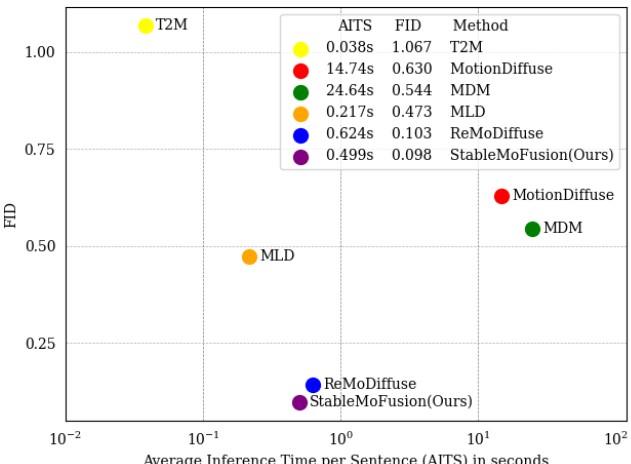

Figure 1: Comparison of the inference time costs on motion generation. The closer the model is to the origin, the better.

Despite the notable progress made by diffusion-based motion generation methods, its development is still hindered by several fragmented and underexplored issues: 1) **Lack of Systematic Analysis**: these diffusion-based motion generation work usually employ different network architectures and training pipelines, which hinders cross-method integration and the adoption of advancements from related domains. 2) **Long Inference Time**: due to the time-consuming iterative sampling process, most existing methods are impractical for applications with virtual characters and humanoid robots, where real-time responsiveness is crucial. 3) **Footskate Issue**: foot skating (footskate) in generated motions remains a major concern. This significantly undermines the quality of generated motions and limits their practical applicability.

Therefore, in order to fill these research gaps and enhance the effectiveness and reliability of diffusion-based motion generation in practical applications, our study conducts a comprehensive and systematic investigation into network architectures, training strategies, and inference process. Our investigation is specifically directed

towards text conditional motion generation, as text prompts are arguably the most promising format for practical application and the most convenient input modality among various conditional signals. Ultimately, we present a robust and efficient framework for diffusion-based motion generation, called *StableMoFusion*, as illustrated in Figure 2.

In StableMoFusion, we use Conv1D UNet with AdaGN and linear cross-attention as the motion-denoising network, and improve its generalization capability with GroupNorm tweak. During training, two effective strategies were employed to enhance the network's ability to generate motion. During inference, we use four training-free acceleration tricks to achieve efficient inference. Furthermore, we present a footskate cleanup method based on a mechanical model and optimization.

Extensive experiments demonstrate that StableMoFusion achieves an excellent trade-off between text-motion consistency and motion quality compared to other state-of-the-art methods, as shown in Table 1. Meanwhile, Stablemofusion's efficient inference process notably reduces the minimum number of iterations required for generation from 1000 to 10, as well as shorter inference times than methods of about the same performance, achieving an average inference time of 0.5 seconds on the Humanm3D test set, as shown in Figure 1. In addition, our footskate cleanup method within diffusion framework sizably solves the foot skating problem of motion generation as shown in Section 5.4.

Our major contributions can be summarized as follows:

- We perform a systematic evaluation and analysis on the design of each component in the diffusion-based motion generation pipeline, including network architectures, training strategies, and inference process.
- We propose an effective mechanism to eliminate foot skating which is a comment issue in current methods.
- By consolidating these well-designed components, we present a robust and efficient diffusion-based motion generation framework named *StableMoFusion*. Extensive experiments demonstrate its superiority in text-motion consistency and motion quality.

## 2 RELATED WORK

### 2.1 Motion Diffusion Generation

In recent years, the application of diffusion models to human motion generation has led to significant improvements in the quality of generated motions. MotionDiffuse [36] softly fuses text features into diffusion-based motion generation through cross-attention. MDM [27] experimented with the separate Transformer encoder, decoder, GRU as denoising networks, respectively. PyhsDiff [34] incorporates physical constraints to generate more realistic motions; Prior MDM [23] uses diffusion priors to allow the model to be applied to specific generative tasks; MLD [3] utilizes the latent space of VAE to speed up diffusion generation; ReMoDiffuse [37] uses a retrieval mechanism to enhance the motion diffusion model. All of these methods use Transformer-based network structure, while MoFusion [4] and GMD [13] use Conv1D UNet for motion diffusion generation.

Our work towards a more robust and efficient diffusion-based motion generation framework through a comprehensive investigation into network architectures, training strategies, and inference process. It also addresses the practical application challenges of long inference time and footskate phenomenon.

### 2.2 Training-Free Sampling

To reduce the inference time with a trained network, there have been many advanced samplers to accelerate DDPM [8].

Song et al. [25] show that using Stochastic Differential Equation (SDE) for sampling has a marginally equivalent probability Ordinary Differential Equations (ODE). And then, DDIM [24] constructs a class of non-Markovian diffusion processes that realize skip-step sampling. PNDN [15] uses pseudo numerical to accelerate the deterministic sampling process. DEIS [38] and DPMSolver [16] improve upon DDIM by numerically approximating the score functions within each discretized time interval.

Meanwhile, several work have focused on speeding up stochastic sampling. For example, Gotta Go Fast [11] utilizes adaptive step sizes to speed up SDE sampling, and Lu et al. [17] converts the higher-order ODE solver into an SDE sampler to address the instability issue.

While these samplers have demonstrated efficacy in image generation, their impact on motion diffusion models remains unexplored. In this work, we evaluate them to find the most appropriate one for motion generation.

### 2.3 Footskate Cleanup

In order to generate realistic motions in computer animation, various methods have been developed to improve footskate issue.

Edge [28] embeds the foot contact term into the action representation for training and applies Contact Consistency Loss as a constraint to keep the physical plausibility of motion. RFC [33], Drop [10] and Physdiff [34] uses reinforcement learning to constrain the physical states of actions, such as ground force reaction and collision situations to get a realism motion. UnderPressure [18] and GroundLink [7] respectively collect foot force datasets during motion. UnderPressure [18] also utilizes this dataset to train a network capable of predicting vertical ground reaction forces. Based on this, UnderPressure proposes a foot skating cleanup method.

## 3 PRELIMINARIES

The pipeline of Diffusion model [8] involve three interconnected processes: a **forward process** that gradually diffuses noise into sample, a **reverse process** that optimizes a network to eliminate the above perturbation, and an **inference process** that utilizes the trained network to iteratively denoise noisy sample.

Specifically, a motion denoising network is first trained to predict the original motion $x_0$ from the noisy motion $x_t$: randomly select a ground-truth motion $x_0$ and a diffusion timestep $t \sim U[0, T]$, $T$ being the maximum timestep. And then the noisy motion $x_t$ after t-step diffusion is gained by Equation 1,

$$x_t = \sqrt{\bar{\alpha}_t} x_0 + \sqrt{1 - \bar{\alpha}_t} \epsilon \tag{1}$$

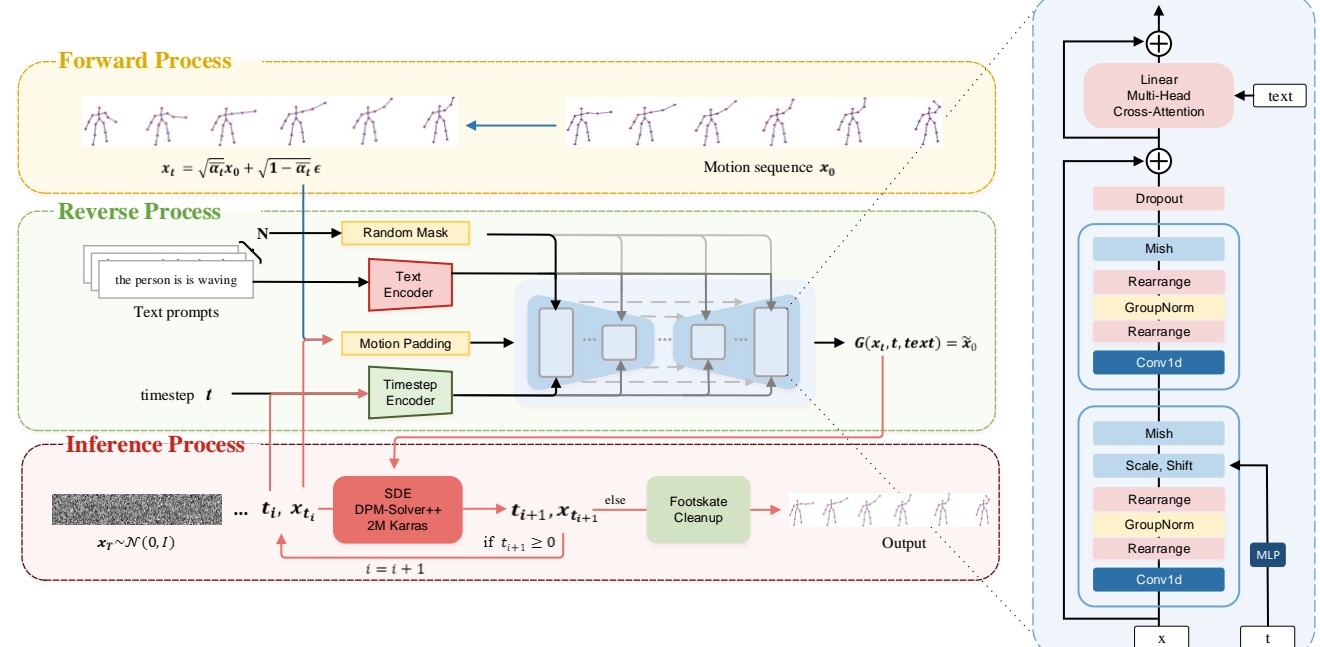

**Figure 2: Overview of StableMoFusion, which is composed of a diffusion forward process, a reverse process on CondUNet1D motion-denoising network, and an efficient inference. The colors of the arrows indicate different stages: blue for training, red for inference, and black for both.**

where $\epsilon$ is a Gaussian noise. $\sqrt{\bar{\alpha}_t}$ and $\sqrt{1-\bar{\alpha}_t}$ are the strengths of signal and noise, respectively. When $\sqrt{\bar{\alpha}_t}$ is small enough, we can approximate $x_t \sim \mathcal{N}(0, I)$.

Next, given a motion-denoising model $G_\theta(x_t, t, c)$ for predicting the original sample, parameterized by $\theta$, the optimization can be formulated as follows:

$$\min_\theta E_{t\sim U[0,T], x_0\sim p_{data}} ||G_\theta(x_t, t, c) - x_0||_2^2 \qquad (2)$$

In the inference process, a trained motion-denoising network can progressively generate samples from noise with various samplers. For instance, DDPM [8] iteratively denoise the noisy data from $t$ to a previous timestep $t'$, as shown in Algorithm 1.

---

**Algorithm 1** Inference

Given a text prompt $c$
$\mathbf{x}_t \sim \mathcal{N}(0, \mathbf{I})$
**for** $t = T$ to $1$ **do**
$\quad \widetilde{\mathbf{x}}_0 = G(\mathbf{x}_t, t, c)$
$\quad \epsilon \sim \mathcal{N}(0, I)$ if $t > 1$, else $\epsilon = 0$
$\quad \mathbf{x}_{t-1} = \frac{\sqrt{\bar{\alpha}_{t-1}}\beta_t}{1-\bar{\alpha}_t}\widetilde{\mathbf{x}}_0 + \frac{\sqrt{\alpha_t}(1-\bar{\alpha}_{t-1})}{1-\bar{\alpha}_t}\mathbf{x}_t + \tilde{\beta}_t\epsilon$
**end for**
**return** $\mathbf{x}_0$

---

## 4 METHOD

Through comprehensive exploratory experiments conducted on diffusion-based motion generation, we propose a novel diffusion

framework, named StableMoFusion, as illustrated in Figure 2, to facilitate robust and efficient motion generation. This section begins with our investigation on the architecture of motion-denoising networks. Next, we discuss several training strategies pivotal for enhancing model performance in Section 4.2. Subsequently, we introduce our improvements in the inference process in Section 4.3, tailored to enable efficient inference. Lastly, we discuss and present a solution to the footskate issue in Section 4.4.

### 4.1 Model Architecture

Most existing work use Transformer [29]-based architectures as the motion-denoising network; however, it remains questionable whether these architectures are best for diffusion-based motion generation. In this subsection, we will present three new network architectures fine-tuned for the motion generation task: Conv1D UNet [4, 13], Diffusion Transformer (DiT) [19] and the latest Retentive Network (RetNet) [26].

#### 4.1.1 Conv1D UNet.

*Baseline.* We chose the Conv1D UNet with AdaGN [5] and skip connections in GMD [13] as the Conv1D UNet baseline and modify the structure to a canonical Unet structure, which consist of four downsampling stages. The motion length $n$ is successively reduced from $N$ to $\lfloor N/8 \rfloor$, and then the corresponding up-sampling phase is used to up-sample. There are two residual Conv1D blocks for each down-sampling or up-sampling stage, with a single block shown as Figure 3 (a).

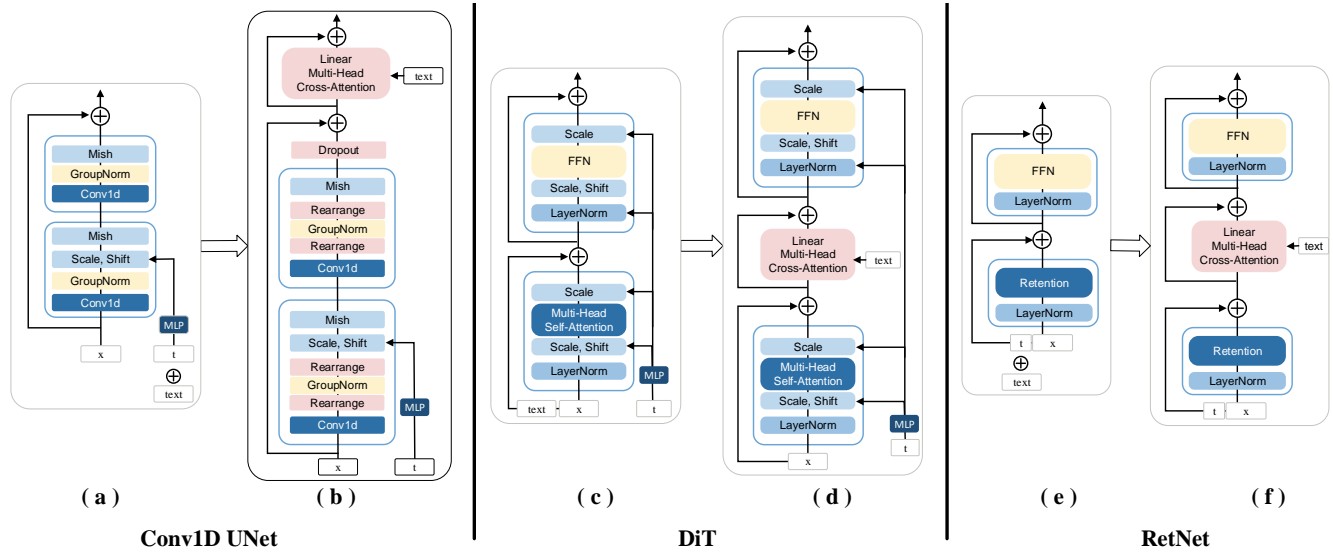

**Figure 3: Visualization of the block structure and their adjustments of Conv1D UNet, DiT and RetNet. Pink blocks indicate structures that have been added or modified.**

*Block Adjustment.* We introduce Residual Linear Multi-Head Cross-Attention after each block to effectively integrate textual cues, and dropout is incorporated into the original Conv1D block to enhance model generalization, as shown in Figure 3 (b). In the baseline block, text prompts are encoded with timesteps and integrated into motion coding using a simple formula: $x \cdot (1 + scale) + shift$. However, this approach doesn't effectively incorporate textual cues into motion sequences because it applies uniform operations across the entire sequence. In diffusion pipelines, noise uniformly affects the entire sequence, resulting in consistent mappings between motion frames and timesteps. However, since each frame's motion corresponds to distinct textual cues, a straightforward "scale and shift" approach is insufficient for injecting textual information. Our solution employs an attention mechanism to dynamically focus each motion frame on its associated textual information. Residual connections help mitigate potential computation biases introduced by cross attention.

*GroupNorm Tweak.* We rearranged the data before and after applying Group Normalization, as depicted in Figure 3 (b), to minimize the impact of padded data during network forward propagation. When testing the adapted Conv1D UNet on datasets like KIT-ML with varying sequence lengths, we noticed a significant performance drop. This suggests that the model struggles with datasets containing extensive padding. Further investigation revealed that implementing Group Normalization within the baseline block caused this issue. Since Conv1D operates along the temporal dimension, directly applying Group Normalization to the input disrupted the differentiation between padded and non-padded data, affecting loss computation and gradient descent.

### 4.1.2 Diffusion Transformer.

*Baseline.* To explore the effectiveness of the DiT structure for motion generation, we replace the Vision Transformer used for

images in the DiT with self-attention used for motion data as the baseline, with the basic block structure shown in Figure 3 (c). For text-to-motion generation, we embed text prompts via the CLIP [21] encoder and project them into token concatenated with motion embeddings for self-attention. It scales and shifts the motion embedding before and after each autoregressive computation using timestep, which ensures the motion denoising trajectory closely aligned with the timestep.

*Block Adjustment.* We have also tried to incorporate Linear Multi-Head Cross-Attention into the DiT framework, as shown in Figure 3 (d). This adjustment allows for a more nuanced fusion of textual cues with motion dynamics than fusing all the text information into the one-dimensional text embedding in baseline, which enhances the coherence and relevance of generated motion sequences.

### 4.1.3 Retentive Network.

*Baseline.* Our RetNet baseline follows a straightforward implementation similar to MDM, where the timesteps encoding is concatenated with the textual projection to form tokens, which are then fed along with motion embeddings into RetNet, with its basic block shown in Figure 3 (e). RetNet incorporates a gated multi-scale retention mechanism, which enhances information retention and processing capabilities, thereby enabling nuanced comprehension and generation of motion sequences. Through our investigation, we aim to ascertain the feasibility of leveraging RetNet for motion generation tasks.

*Block Adjustment.* To further integrate textual information, we also employ Linear Multi-Head Cross-Attention between retention and FFN, as shown in Figure 3 (f). By segregating temporal and textual features, our approach aims to preserve the distinct characteristics of each modality and allow the model to independently

learn and leverage relevant cues for motion generation. This separation enhances the model's interpretability and flexibility, enabling it to better capture the intricacies of both temporal dynamics and semantic context.

### 4.1.4 Final Model Architecture.

Ultimately, we choose the Conv1D UNet with block adjustment and GroupNorm tweak as the motion-denoising model of StableMoFusion, as shown in Figure 2. We call this network as CondUNet1D. Both DiT and RetNet use the idea of attention to activate the global receptive field in the temporal dimension, which benefits the modeling of long-range dependency. The receptive field of Conv1D UNet is mainly in the convolution kernel window, promoting a coherent and smooth transition between frames. We tend to prioritize smoother generation in current applications of motion generation.

In our StableMoFusion, we set the base channel and channel multipliers of UNet to 512 and [2,2,2,2] respectively. For text encoder, we leverage pre-trained CLIP [21] token embeddings, augmenting them with four additional transformer encoder layers, the same as MotionDiffuse [36], with a latent text dimension of 256. For timesteps encoder, it is implemented using position encoding and two linear layers, the same as StableDiffusion [22], with a latent time dimension of 512.

## 4.2 Training strategies

Recent research has shown that key factors in the training strategies of the diffusion model affect the learning pattern and its generative performance [2]. In this subsection, we will analyze the impact of two empirically valid training strategies on diffusion-based motion generation: exponential moving average and classifier-free guidance.

### 4.2.1 Exponential Moving Average.

Exponential Moving Average (EMA) calculates a weighted average of a series of model weights, giving more weight to recent data. Specifically, assume the weight of the model at time t as $\theta_t$, then the EMA formulated as: $v_t = \beta \cdot v_{t-1} + (1 - \beta) \cdot \theta_t$, where $v_t$ denotes the average of the network parameters for the first t iterations ($v_0 = 0$), and $\beta$ is the weighted weight value.

During the training of the motion-denoising network, the network parameters change with each iteration, and the motion modeling oscillates between text-motion consistency and motion quality. Therefore, the use of EMA can smooth out the change process of these parameters, reduce mutations and oscillations, and help to improve the stability ability of the motion-denoising model.

### 4.2.2 Classifier-Free Guidance.

To further improve the generation quality, we use Classifier-Free Guidance (CFG) to train the motion-denoising generative model. By training the model to learn both conditioned and unconditioned distributions (e.g., setting c = ∅ for 10% of the samples), CFG ensures that the models can effectively capture the underlying data distribution across various conditions. In inference, we can trade-off text-motion consistency and fidelity using s:

$$G_s(x_t, t, c) = G(x_t, t, \emptyset) + s \cdot (G(x_t, t, c) - G(x_t, t, \emptyset)) \quad (3)$$

This ability to balance text-motion consistency and fidelity is crucial for producing varied yet realistic outputs, enhancing the overall quality of generated motion.

## 4.3 Efficient Inference

Time-consuming inference time remains a major challenge for diffusion-based approaches. To address this problem, we improve inference speed by integrating four efficient and training-free tricks in the inference process: 1) efficient sampler, 2) embedded-text cache, 3) parallel CFG computation, and 4) low-precision inference.

### 4.3.1 Efficient Sampler.

We integrate the SDE variant of second-order DPM-Solver++ sampler (SDE DPM-Solver++ 2M) into diffusion-based motion generation to reduce denoising iterations. DPM-Solver is a high-order solver for diffusion stochastic differential equations (SDEs), which implies additional noise will be introduced during the iterative sampling. Thereby, stochasticity of its sampling trajectories helps to reduce the cumulative error [32], which is crucial for the realism of generated motion. In addition, we adopt the Karras Sigma [12] to set discrete timesteps. This method leverages the theory of constant-velocity thermal diffusion to determine optimal timesteps, thereby maximizing the efficiency of motion denoising within a given number of iterations.

### 4.3.2 Embedded-text Cache.

We integrate the Embedded-text Cache mechanism into the inference process to avoid redundant calculations. In diffusion-based motion generation, the text prompt remain unchanged across iterations, resulting in same embedded text in each computation of the denoising network. Specifically, we compute the text embedding initially and subsequently utilize the embedded text directly in each network forward, thereby reducing computational redundancy and speeding up inference.

### 4.3.3 Parallel CFG Computation.

We implement the inference process of CFG in parallel to speed up the single iteration calculation while maintaining model generation performance. Due to the CFG mechanism Equation 3, in each iterative step during inference, it is necessary to execute a conditional and an unconditional denoising, respectively, using the trained motion network, and then sum up the results.

### 4.3.4 Low-precision Inference.

We utilize half-precision floating point (FP16) computation during inference to accelerate processing. Newer hardware supports enhanced arithmetic logic units for lower-precision data types. By applying parameter quantization, we convert FP32 computations to lower-precision formats, effectively reducing computational demands, parameter size, and memory usage of the model.

## 4.4 Footskate Reduction

Figure 4 shows an example for the foot skating phenomenon. The motion frame rate is 20. The two frames in the figure have a time difference of 0.25s. Based on our life experience, it is difficult to complete a motion and return to the original pose within 0.25s. Although the foot postures in the two frames remain unchanged, there are changes in the positions of the joints, as observed from

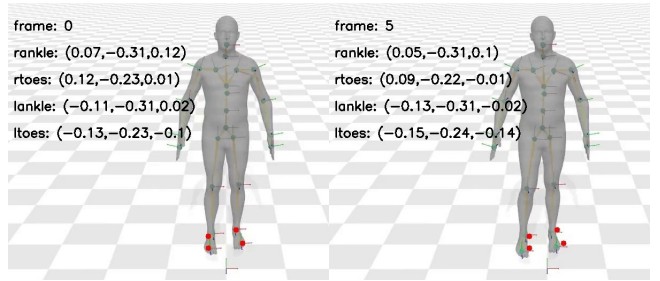

**Figure 4: Red: the foot joints as 0th frame. Green: the corresponding keypoints. At 5th frame, the offset of red and green points indicate the footskate phenomenon.**

the variations in joint position values and their distances relative to red points. For this motion, what we expect is that the feet are anchored at the same point. Typically, choosing the foot position of middle frames during foot skating as the fixed point minimizes the impact on the adjacent frames.

The key to eliminating foot skating is to first identify the foot joints and frame ranges where foot skating occurs, and then anchor those keypoints at their positions $p$ in the intermediate frames. We formulate this constraint as a loss term shown in Equation 4 where $j$ indicates joint and $f$ is frame ranges.

$$L_{foot} = \sum_{j}^{J_{skating}} \sum_{f}^{F_{skating}} (P_j - p) \tag{4}$$

$J_{skating}$ contains all the joints where foot skating may occur, specifically including right ankle, right toes, left ankle and left toes. $F_{skating}$ is a collection of all frame ranges of the joint $j$. $P_j$ means the positions of joint $j$. We incorporate Equation 4 to a gradient descent algorithm to correct foot skating motion.

Following UnderPressure [18], we use vertical ground reaction forces (vGRFs) to identity foot joint $j$ and its skating frames $f$. The vGRFs predition model of UnderPressure $V_{23}$ requires motion of a predefined 23 joints skeleton $S_{23}$, which is different from our motion data. In our work, we utilize HumanML3D[6] with 22 skeletal joints $S_{22}$ and KIT-ML [20] motion with 21 skeletal joints. The subsequent foot skating cleanup primarily focused on HumanML3D. We transferred the pre-trained weights of $V_{23}$ to our own model $V_{22}^{\theta}$ using the constraints Equation 5, enabling us to directly predict the vertical ground reaction forces for HumanML3D motions. $P$ is keypoints of HumanML3D motion. $P_{S_{23}}$ is the result of retargeting $P$ to skeleton $S_{23}$.

$$\min_{\theta} \|V_{22}^{\theta}(P) - V_{23}(P_{S_{23}})\|_2^2 \tag{5}$$

$$L = \omega_q L_{pose} + \omega_f L_{foot} + \omega_t L_{trajectory} + \omega_v L_{vGRFs} \tag{6}$$

$$L_{foot} = L_{foot}(P, \hat{P}, V_{23}, P_{S_{23}}) \tag{7}$$

$$L_{vGRFs} = L_{vGRFs}(P, \hat{P}, V_{22}^{\theta}) \tag{8}$$

Drawing inspiration from UnderPressure [18], we use foot contact loss $L_{foot}$ to fix contact joints, pose loss $L_{pose}$ and trajectory loss $L_{trajectory}$ to to keep the semantic integrity of motion, vGRFs loss $L_{vGRFs}$ to keep valid foot pose. Our supplementary material

provides detailed definitions of these loss terms. The final definition of our loss function is as Equation 6 [18] where $\omega_q$, $\omega_f$, $\omega_t$, $\omega_v$ are weights of its loss item. $P$ is keypoints of footskating motion and $\hat{P}$ is the result keypoints after footskate cleanup.

Through our method, the footskate cleanup process can be generalized to various skeletal motions.

In a few cases, motion corrected by Equation 6 may occurs unreasonable or unrealistic poses. The diffusion model trained on a large amount of motion data learns the prior knowledge of real motions and has the ability to correct the invalid motions.

Therefore, we use our pretrained diffusion model to correct such cases. Motivated by OmniControl [31] and Physdiff [34], we incorporates footskate cleaning method into the diffusion denoising process, denote as StableMoFusion*.

## 5 EXPERIMENTS

### 5.1 Dataset and Evaluation Metrics

We use HumanML3D [6] and KIT-ML [20] dataset for our experiments. HumanML3D Dataset contains 14,646 motions and 44,970 motion annotations. KIT Motion Language Dataset contains 3,911 motions and 6,363 natural language annotations.

The evaluation metrics can be summarized into four key aspects: 1) Motion Realism: Frechet Inception Distance (FID), which evaluates the similarity between generated and ground truth motion sequences using feature vectors extracted by a pre-trained motion encoder [6]. 2) Text match: R Precision calculates the average top-k accuracy of matching generated motions with textual descriptions using a pre-trained contrastive model [6]. 3) Generation diversity: Diversity measures the average joint differences across generated sequences from all test texts. Multi-Modality quantifies the diversity within motions generated for the same text. 4) Time costs: Average Inference Time per Sentence (AITS) [3] measures the inference efficiency of diffusion models in seconds, considering generation batch size as 1, without accounting for model or data loading time.

In all of our experiments, FID and R Precision are the principal metrics we used to conduct our analysis and draw conclusions.

### 5.2 Implements Details

For training, we use DDPM [8] with $T = 1,000$ denoising steps and variances $\beta_t$ linearly from 0.0001 to 0.02 in the forward process. And we use AdamW with an initial learning rate of 0.0002 and a 0.01 weight decay to train the sample-prediction model for 50,000 iterations at batch size 64 on an RTX A100. Meanwhile, learning rate reduced by 0.9 per 5,000 steps. On gradient descent, clip the gradient norm to 1. For CFG, setting c = ∅ for 10% of the samples.

For inference, we use the SDE variant of second-order DPM-Solver++ [17] with Karras Sigmas [12] in inference for sampling 10 steps. The scale for CFG is set to 2.5.

### 5.3 Quantitative results

We compare our StableMoFusion with several state-of-the-art models, including T2M [6], MDM [27], MLD [3], MotionDiffuse [36], T2M-GPT [35], MotionGPT [9], ReMoDiffuse [37], M2DM [14] and fg-T2M [30]. on the HumanML3D [6] and KIT-ML [20] datasets in Table 2 and Table 3, respectively. Most results are borrowed from

**Table 2: Quantitative results on the HumanML3D test set. The right arrow → means the closer to real motion the better. Red and Blue indicate the best and the second best result.**

| Method | FID ↓ | R Precision↑ | | | Diversity → | Multi-modality ↑ |
|--------|-------|------|------|------|-------------|------------------|
| | | top1 | top2 | top3 | | |
| Real | $0.002^{\pm.000}$ | $0.511^{\pm.003}$ | $0.703^{\pm.003}$ | $0.797^{\pm.002}$ | $9.503^{\pm.065}$ | - |
| T2M [6] | $1.067^{\pm.002}$ | $0.457^{\pm.002}$ | $0.639^{\pm.003}$ | $0.743^{\pm.003}$ | $9.188^{\pm.002}$ | $2.090^{\pm.083}$ |
| MDM [27] | $0.544^{\pm.044}$ | $0.320^{\pm.005}$ | $0.498^{\pm.004}$ | $0.611^{\pm.007}$ | $\color{blue}9.599^{\pm.086}$ | $\color{blue}2.799^{\pm.072}$ |
| MLD [3] | $0.473^{\pm.013}$ | $0.481^{\pm.003}$ | $0.673^{\pm.003}$ | $0.772^{\pm.002}$ | $9.724^{\pm.082}$ | $2.413^{\pm.079}$ |
| MotionDiffuse [36] | $0.630^{\pm.001}$ | $0.491^{\pm.001}$ | $0.681^{\pm.001}$ | $0.782^{\pm.001}$ | $9.410^{\pm.049}$ | $1.553^{\pm.042}$ |
| GMD [13] | 0.212 | - | - | 0.670 | 9.440 | - |
| T2M-GPT [35] | $0.116^{\pm.004}$ | $0.491^{\pm.003}$ | $0.680^{\pm.003}$ | $0.775^{\pm.002}$ | $9.761^{\pm.081}$ | $1.856^{\pm.011}$ |
| MotionGPT [9] | $0.232^{\pm.008}$ | $0.492^{\pm.003}$ | $0.681^{\pm.003}$ | $0.778^{\pm.002}$ | $\color{red}9.528^{\pm.071}$ | $2.008^{\pm.084}$ |
| ReMoDiffuse [37] | $\color{blue}0.103^{\pm.004}$ | $\color{blue}0.510^{\pm.005}$ | $\color{blue}0.698^{\pm.006}$ | $\color{blue}0.795^{\pm.004}$ | $9.018^{\pm.075}$ | $1.795^{\pm.043}$ |
| M2DM [14] | $0.352^{\pm.005}$ | $0.497^{\pm.003}$ | $0.682^{\pm.002}$ | $0.763^{\pm.003}$ | $9.926^{\pm.073}$ | $\color{blue}3.587^{\pm.072}$ |
| Fg-T2M [30] | $0.243^{\pm.019}$ | $0.492^{\pm.002}$ | $0.683^{\pm.003}$ | $0.783^{\pm.002}$ | $9.278^{\pm.072}$ | $1.614^{\pm.049}$ |
| StableMoFusion (Ours) | $\color{red}0.098^{\pm.003}$ | $\color{red}0.553^{\pm.003}$ | $\color{red}0.748^{\pm.002}$ | $\color{red}0.841^{\pm.002}$ | $9.748^{\pm.092}$ | $1.774^{\pm.051}$ |

**Table 3: Quantitative results on the KIT-ML test set. The right arrow → means the closer to real motion the better. Red and Blue indicate the best and the second best result.**

| Method | FID ↓ | R Precision↑ | | | Diversity → | Multi-modality ↑ |
|--------|-------|------|------|------|-------------|------------------|
| | | top1 | top2 | top3 | | |
| Real Motion | $0.031^{\pm.004}$ | $0.424^{\pm.005}$ | $0.649^{\pm.006}$ | $0.779^{\pm.006}$ | $11.08^{\pm.097}$ | - |
| T2M [6] | $2.770^{\pm.109}$ | $0.370^{\pm.005}$ | $0.569^{\pm.007}$ | $0.693^{\pm.007}$ | $10.91^{\pm.119}$ | $1.482^{\pm.065}$ |
| MDM [27] | $0.497^{\pm.021}$ | $0.164^{\pm.004}$ | $0.291^{\pm.004}$ | $0.396^{\pm.004}$ | $10.847^{\pm.109}$ | $1.907^{\pm.214}$ |
| MLD [3] | $0.404^{\pm.027}$ | $0.390^{\pm.008}$ | $0.609^{\pm.008}$ | $3.204^{\pm.027}$ | $10.80^{\pm.117}$ | $2.192^{\pm.071}$ |
| MotionDiffuse [36] | $1.954^{\pm.062}$ | $0.417^{\pm.004}$ | $0.621^{\pm.004}$ | $0.739^{\pm.004}$ | $\color{red}11.10^{\pm.143}$ | $0.730^{\pm.013}$ |
| T2M-GPT [35] | $0.514^{\pm.029}$ | $0.416^{\pm.006}$ | $0.627^{\pm.006}$ | $0.745^{\pm.006}$ | $10.921^{\pm.108}$ | $1.570^{\pm.039}$ |
| MotionGPT [9] | $0.510^{\pm.016}$ | $0.366^{\pm.005}$ | $0.558^{\pm.004}$ | $0.680^{\pm.005}$ | $10.35^{\pm.084}$ | $\color{blue}2.328^{\pm.117}$ |
| ReMoDiffuse [37] | $\color{red}0.155^{\pm.006}$ | $\color{blue}0.427^{\pm.014}$ | $\color{blue}0.641^{\pm.004}$ | $\color{blue}0.765^{\pm.055}$ | $10.80^{\pm.105}$ | $1.239^{\pm.028}$ |
| M2DM [14] | $0.515^{\pm.029}$ | $0.416^{\pm.004}$ | $0.628^{\pm.004}$ | $0.743^{\pm.004}$ | $11.417^{\pm.97}$ | $\color{red}3.325^{\pm.37}$ |
| Fg-T2M [30] | $0.571^{\pm.047}$ | $0.418^{\pm.005}$ | $0.626^{\pm.004}$ | $0.745^{\pm.004}$ | $10.93^{\pm.083}$ | $1.019^{\pm.029}$ |
| StableMoFusion (Ours) | $\color{blue}0.258^{\pm.029}$ | $\color{red}0.445^{\pm.006}$ | $\color{red}0.660^{\pm.005}$ | $\color{red}0.782^{\pm.004}$ | $\color{blue}10.936^{\pm.077}$ | $1.362^{\pm.062}$ |

their own paper and we run the evaluation 20 times and ± indicates the 95% confidence interval.

Our method achieves the state-of-the-art results in FID and R Precision (top k) on the HumanML3D dataset, and also achieves good results on the KIT-ML dataset: the best R Precision (top k) and the second best FID. This demonstrates the ability of StableMoFusion to generate high-quality motions that align with the text prompts. On the other hand, while some methods excel in diversity and multi-modality, it's crucial to anchor these aspects with accuracy (R-precision) and precision (FID) to strengthen their persuasiveness. Otherwise, diversity or multimodality becomes meaningless if the generated motion is bad. Therefore, our StableMoFusion achieves advanced experimental results on two datasets and shows robustness in terms of model performance.

StableMoFusion* focuses on the real effect of footskate cleanup. Therefore, the timestep to begin cleaning footskate during inference depends on the motion and thus the StableMoFusion* doesn't apply to the evaluation process of [6].

## 5.4 Qualitative result

Figure 5 shows the visual results of our footskate cleanup method, StableMoFusion*. The red bounding box of footskate motion clearly has multiple foot outlines, whereas ours shows only one. The comparison graph shows the effectiveness of our method for cleaning footskate. Directly applying the footskate cleanup method of UnderPressure [18] to our motion would result in motion distortion, while our method effectively avoids such deformation. In our supplementary material, we will further present a comparison between our method and the UnderPressure method by videos to illustrate it.

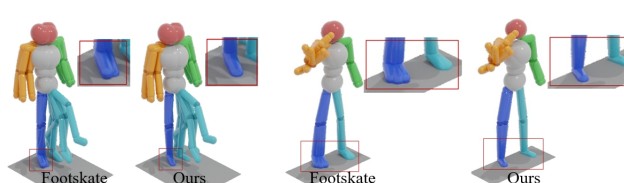

Caption: kick left leg step back        Caption: a person waves with his right hand

**Figure 5: Visualization comparison results before and after our footskate cleanup. The red bounding box shows details of skating feet.**

## 5.5 Inference Time

We calculate AITS of StableMoFusion and ReMoDiffuse [37] with the test set of HumanML3D[6] on Tesla V100 as MLD [3] does, the other results of Figure 1 are borrowed from [3]. For instance, MDM [27] with CFG requires 24.74s for average inference; MotionDiffuse [36] without CFG uses condition encoding cache and still requires 14.74s of average inference. While the MLD [3] reduces the average inference time to 0.217s by applying DDIM50 in latent space, we find this approach lacks the ability to edit and control motion by manipulating the model input.

To tackle this, we employ 1) efficient sampler, 2) embedded-text cache, 3) parallel CFG computation, and 4) low-precision inference to reduce iteration counts and network latency. As shown in Figure 1, our StableMoFusion significantly shortens the inference time and achieves higher performance within the original motion space.

However, it remains incontrovertible that StableMoFusion's inference speed trails behind that of MLD, and fails to meet the industry's real-time standard with an average inference time of 0.5s. Thus, our future work will focus on acceleration: the inference time of StableMoFusion is currently tied to the computation of the network, and we will further investigate how to scale down the model and how to reduce single-step latency in inference.

## 5.6 Ablation

### 5.6.1 Network Architecture.
We evaluate and compare all the architectures mentioned in Section 4.1 with the same training and inference pipeline. For a fair comparison, all methods use the real motion length from the ground truth to clip generated motion and seed(0) for one evaluation. As Table 4 show, each network enhancement in cross-attention has demonstrated performance enhancements, elucidating its pivotal role in augmenting model

efficacy and effectiveness. Among them, Conv1D UNet achieves the best generation performance. And fine-tuning Conv1D UNet's GroupNorm can effectively improve its performance on the KIT-ML dataset, reducing the FID by about 64%. It also proves that the GoupNorm tweak on UNet is mainly useful for the dataset with dispersed length distributions, such as KIT-ML dataset.

**Table 4: Comparison of various architectures and adjustments.**

| Dataset | Network | FID ↓ | R Precision (top3) ↑ |
|---|---|---|---|
| HumanML3D | Conv1D UNet basline | 0.245 | 0.780 |
| | + cross-attention | 0.074 | 0.821 |
| | + GroupNorm Tweak | 0.089 | 0.840 |
| | DiT baseline | 0.884 | 0.711 |
| | + cross-attention | 0.113 | 0.787 |
| | RetNet baseline | 1.673 | 0.740 |
| | + cross-attention | 0.147 | 0.853 |
| KIT-ML | Conv1D UNet+ cross-attention | 0.658 | 0.756 |
| | + GroupNorm Tweak | 0.237 | 0.780 |

### 5.6.2 Effectie Inference.
By using the SDE variant of second-order DPM-Solver++ with Karras sigma, the inference process of diffusion-based motion generation is able to significantly reduce the minimum number of iterations required for generation from 1000 to 10 while enhancing the quality of generated motions, approximately 99% faster than the original inference process, as shown in Table 5.

The application of embedded text caching and parallel CFG further reduces the average inference time by about 0.3s and 0.15s, respectively. Our experiments also show that reducing the computational accuracy of the motion-denoising model by half, from FP32 to FP16, does not adversely affect the generation quality. This suggests that 32-bit precision is redundant for motion generation task.

**Table 5: The progressive effect of each efficient and training-free trick of StableMoFusion in inference process.**

| Method | FID↓ | R Precision (top3)↑ | AITS↓ | Inference Steps↓ |
|---|---|---|---|---|
| base (DDPM1000) | 1.251 | 0.760 | 99.060 | 1000 |
| + Efficient Sampler | 0.076 | 0.836 | 1.004(-99%) | 10 |
| + Embedded-text Cache | 0.076 | 0.836 | 0.690(-31%) | 10 |
| + Parallel CFG | 0.076 | 0.836 | 0.544(-21%) | 10 |
| + FP16 | 0.076 | 0.837 | 0.499(-8%) | 10 |

## 6 CONCLUSION

In this paper, we propose a robust and efficient diffusion-based motion generation framework, StableMoFusion, which uses Conv1DUNet as a motion-denoising network and employs two effective training strategies to enhance the network's effectiveness, as well as four training-free tricks to achieve efficient inference. Extensive experimental results show that our StableMoFusion performs favorably against current state-of-the-art methods. Furthermore, we propose effective solutions for time-consuming inference and footskate problems, facilitating diffusion-based motion generation methods for practical applications in industry.

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
