# OpenReview forum: "StableMoFusion: Towards Robust and Efficient Diffusion-based Motion Generation Framework"
_acmmm.org/ACMMM/2024/Conference — MM2024 Oral_

### Official Review · Reviewer_pGnt · 2024-04-30

**Rating:** 6
**Confidence:** 2

**Summary:**

The paper discusses a novel framework for generating human motions using diffusion models. It aims to address the challenges of computational overhead and the common issue of footskate in generated motions, which are prevalent in existing diffusion-based solutions. The authors conduct a comprehensive investigation into network architectures, training strategies, and inference processes to tailor components for efficient and high-quality human motion generation. They also present a method to eliminate footskate by identifying foot-ground contact and correcting foot motions during the denoising process. The framework, StableMoFusion, combines these components to achieve robust and efficient human motion generation, outperforming current state-of-the-art methods in terms of motion quality and inference efficiency.

**Strengths:**

1. I am very pleased with the author's selection of three highly representative structures for comparison, as depicted in Figure 3: the Convolutional U-Net, DiT (Diffusion Transformer), and RetNet (Retentive Network). The comparison is very clear and also very valuable.

2. The ablation study is quite persuasive, and I am very convinced that this is an article worth reading by more people.

3. I interpret the design of the Footskate section as a kind of smoothing, which appears to be effective.

**Limitations:**

please provide a detailed description of where the results for each experimental subject in the ablation study were sourced from

**Suitability:**

2

---

### Official Review · Reviewer_dazf · 2024-05-16

**Rating:** 2
**Confidence:** 3

**Summary:**

This paper is about the text-to-human motion generation task, which is based on the diffusion model. The author conducted a comprehensive investigation of the network architectures, training strategies, and inference processes. In addition, the author describes the challenges in this task, and tries to alleviate the challenges faced in the existing diffusion-based model for this task.

**Strengths:**

1. Investigating the effect of the design of each component in diffusion-based methods.
2. Analysing the challenges in the existing text to human motion generation task.

**Limitations:**

1. The novelty of this paper is limited. For example, the backbone of the diffusion model is generally tested in ablation studies. That is not necessarily a novelty to contribute to this paper.
2. The author mentioned that the proposed method accelerates the inference time of the diffusion model, however, the author only describes the algorithm of the vanilla DDPM in the paper, not listing the comparison or the improved part of the algorithm.
3. The foot skating phenomenon can be seen as a loss function for this task, and other papers also try to alleviate this issue in their papers, but do not consider this as a main contribution. So, the contribution is limited from this point.

**Suitability:**

2

---

### Official Review · Reviewer_hj8j · 2024-06-01

**Rating:** 5
**Confidence:** 3

**Summary:**

This paper tackles the problem of text condtioned human motion generation using diffusion models. The authors conduct a study of relevant architechture details around designing of the overall diffusion-based pipeline, and motivate their changes in comparison to exisiting methods. The primary contribution is in analysis with previous works and introducing consistency terms to reduce the issue of foot skating in generated motions. Experiments include detailed comparisons and ablation study to motivate design decisions and improved performance both in terms of perceptual quality and inference speed.

**Strengths:**

- Detailed comparison study in experiment section with quantitative results across numerous related methods and, comparing results on two different datasets for evaluation. Inclusion of ablation study for some model design choices.

- Method focuses on practical implications like inference speed and perceptual issues with footskate reduction. Althought the method for footskate reduction is inspired from a previous method, certain tweaks on it seem to improve the results.

- The paper seems well-written and sufficiently well-motivated. The figures and supplmentary video were also clear in their presentation.

**Limitations:**

- In terms of analysis for the model architecture which is presented as a contribution, I feel that the study needs some tuning w.r.t choices, like for example transformer v/s unet model. Readers might appreciate if the authors can bring out details like why unet works well (compare with same model but transformer backbone). The current analysis in the paper is definitely not insufficient, but perhaps a different figure in place of Fig 3 could help (current Fig. 3 could go to supplementary as it seems more like an implementation detail).

- Some quantitive measures on the footskate reduction would help showcase the contribution from the method described in Section 4.4 and Appendix A. I assume identification of foot-skating and it's deviation from some ideal behaviour can be done offline? Results on the deviation in the position of foot joints could be a start perhaps.

- Another aspect of footskate reduction that might be worthwhile to consider (and evaluate) is the impact on other joints deviation (from an ideal / or some baseline generation) when enforcing footskate loss. From a couple of videos it seemed that the knees bend more / have a slight jerk to compensate for foot joints remaining static?

**Suitability:**

3

---

### Meta-Review · Area_Chair_4f3f · 2024-06-30

**Recommendation:** Accept (Oral)
**Confidence:** 2

**Metareview:**

The submitted paper presents a comprehensive and well-executed study on human motion generation using a diffusion-based method. The paper's strengths lie in its detailed comparative analysis, practical implications, and robust experimental validation. Despite a few areas that could benefit from further elaboration and clarification, the overall contribution and quality of the work warrant acceptance.

Strengths:

The paper provides a thorough comparison study with quantitative results across numerous related methods. It evaluates performance on two different datasets and includes an ablation study for some model design choices, enhancing the credibility of the results.
The method addresses practical issues such as inference speed and perceptual problems like footskate reduction. Although the method for footskate reduction is inspired by a previous method, certain tweaks improve the results, demonstrating practical relevance and innovation.
The paper is well-written and well-motivated, with clear figures and a supplementary video that effectively present the findings.
The investigation into the design of each component in diffusion-based methods and the challenges in text-to-human motion generation is insightful and valuable.
The ablation study is convincing, demonstrating the importance of various design choices and the effectiveness of the proposed method. This is a significant strength, as it provides clear evidence of the method’s value.
The selection of three highly representative structures for comparison (Convolutional U-Net, DiT (Diffusion Transformer), and RetNet (Retentive Network)) is well thought out and valuable for the community. The comparison is clear and informative.
The footskate section, designed as a form of smoothing, appears to be effective and contributes positively to the quality of the generated motion.

Limitations:

The analysis of the model architecture, specifically the choice between transformer and U-Net models, could be more detailed. Providing more insight into why U-Net works well and comparing it with a transformer backbone would be beneficial. The current analysis is good but could be enhanced, potentially by revising Figure 3 or moving it to the supplementary section.
Including quantitative measures for footskate reduction would strengthen the paper. Specifically, evaluating the deviation in the position of foot joints and the impact on other joints when enforcing footskate loss would provide a more comprehensive analysis.
While the efficient diffusion framework for motion generation is emphasized, the backbone does not necessarily facilitate the inference process, and DPM-Solver++ is an existing method. Thus, the novelty might appear limited. Clarifying the unique contributions and integration would help.
The inference process of the proposed method is not illustrated in the paper. Including this, especially since Algorithm 1 only covers the vanilla DDPM, is important for understanding the efficient diffusion method proposed.
Providing a detailed description of where the results for each experimental subject in the ablation study were sourced from would improve transparency and replicability.


The paper presents a significant contribution to the field of text-to-human motion generation, with a well-motivated approach, thorough comparative analysis, and effective practical solutions. While there are some areas that could be further elaborated, the strengths and overall quality of the work strongly support its acceptance. The authors are encouraged to address the identified limitations in the final revision to further enhance the paper’s clarity and impact.